# Indirect H$_2$O$_2$ synthesis without H$_2$

Arthur G. Fink [1], Roxanna S. Delima[2,3], Alexandra R. Rousseau[3], Camden Hunt[1,2], Natalie E. LeSage[1], Aoxue Huang[1], Monika Stolar[1] & Curtis P. Berlinguette [1,2,3,4] ✉

Industrial hydrogen peroxide (H$_2$O$_2$) is synthesized using carbon-intensive H$_2$ gas production and purification, anthraquinone hydrogenation, and anthrahydroquinone oxidation. Electrochemical hydrogenation (ECH) of anthraquinones offers a carbon-neutral alternative for generating H$_2$O$_2$ using renewable electricity and water instead of H$_2$ gas. However, the H$_2$O$_2$ formation rates associated with ECH are too low for commercialization. We report here that a membrane reactor enabled us to electrochemically hydrogenate anthraquinone (0.25 molar) with a current efficiency of 70% at current densities of 100 milliamperes per square centimeter. We also demonstrate continuous H$_2$O$_2$ synthesis from the hydrogenated anthraquinones over the course of 48 h. This study presents a fast rate of electrochemically-driven anthraquinone hydrogenation (1.32 ± 0.14 millimoles per hour normalized per centimeter squared of geometric surface of electrode), and provides a pathway toward carbon-neutral H$_2$O$_2$ synthesis.

Hydrogen peroxide (H$_2$O$_2$) is an industrial oxidant produced at the megatonne scale (4.3 Mt y$^{-1}$) through the Riedl-Pfleiderer process[1–3]. This process, which accounts for >95% of global H$_2$O$_2$ production[3], involves the reaction of anthraquinone with H$_2$ gas at a Pd catalyst to form anthrahydroquinone (Eq. 1). Anthrahydroquinone then reacts with O$_2$ gas to regenerate anthraquinone and produce H$_2$O$_2$ (Eq. 2)[3,4]. The Riedl-Pfleiderer process relies on H$_2$ gas derived from steam-methane reformation[3]. Steam-methane reformation is an endothermic reaction that produces CO and H$_2$ (1:3 molar ratio) at high temperatures (Eq. 3). This CO is further converted into H$_2$ and CO$_2$ (1:1 molar ratio) in a separate exothermic water-gas shift reaction (Fig. 1A; Eq. 4)[5].

$$\text{Anthraquinone hydrogenation : Anthraquinone} + H_2 \rightarrow \text{Anthrahydroquinone} \quad (1)$$

$$\text{Anthrahydroquinone oxidation : Anthrahydroquinone} + O_2 \rightarrow \text{Anthraquinone} + H_2O_2 \quad (2)$$

$$\text{Steam-methane reformation : } CH_4 + H_2O \rightarrow CO + 3H_2 \quad (3)$$

$$\text{Water-gas shift reaction : } H_2O + CO \rightarrow CO_2 + H_2 \quad (4)$$

Steam-methane reforming associated with the Riedl-Pfleiderer process produces 0.25 mol of CO$_2$ for every mol of H$_2$O$_2$ (i.e., 1.3 Mt$_{CO_2}$ y$^{-1}$) and demands 8.6 gigawatts of energy[6,7]. The electrochemical hydrogenation (ECH) of anthraquinone could bypass such H$_2$ production pathways by instead using water as a hydrogen source[6,8]. With ECH, water oxidation at an anode produces protons that are reduced to reactive hydrogen at a cathode. Reactive hydrogen can subsequently hydrogenation anthraquinone (Supplementary Fig. 1E). The challenge of using ECH is that the heavy aromatic solvents required to dissolve anthraquinone are not compatible with the aqueous medium required for water electrolysis at high current densities[9,10]. Many have attempted to address this solvent incompatibility by (i) immobilizing anthraquinones on the cathode[11–15], (ii) phase-transfer catalysis[6], or (iii) using organic solvent emulsions in alkaline aqueous phase[16,17]. There are also studies showing that H$_2$O$_2$ can be produced directly by ECH[8,18–20]; however, these studies require H$_2$ or suffer from poor electrolytic performance. A summary of direct and indirect processes

[1]Department of Chemistry, The University of British Columbia, 2036 Main Mall, Vancouver, BC V6T 1Z1, Canada. [2]Stewart Blusson Quantum Matter Institute, The University of British Columbia, 2355 East Mall, Vancouver, BC V6T 1Z4, Canada. [3]Department of Chemical and Biological Engineering, The University of British Columbia, 2360 East Mall, Vancouver, BC V6T 1Z3, Canada. [4]Canadian Institute for Advanced Research (CIFAR), 661 University Avenue, Toronto, ON M5G 1M1, Canada. ✉e-mail: cberling@chem.ubc.ca

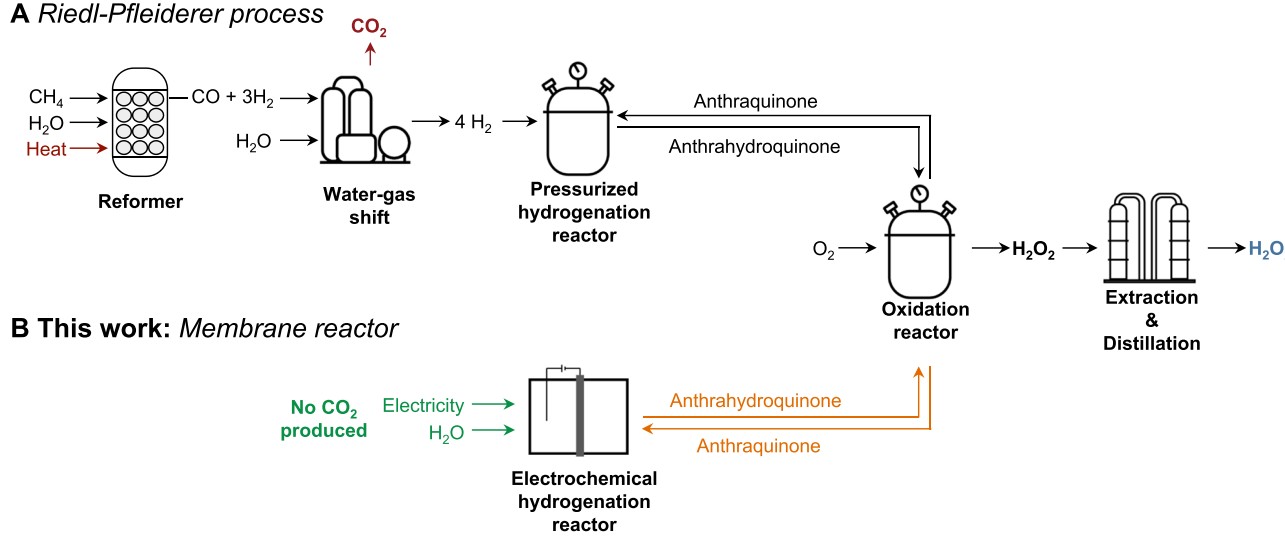

**Fig. 1 | H₂O₂ synthesis. A** Industrial synthesis of H₂O₂ through the Riedl-Pfleiderer process where H₂ is sourced from steam-methane reformation. **B** Electrochemical hydrogenation of anthraquinones where renewable electricity is used to produce reactive hydrogen atoms from water.

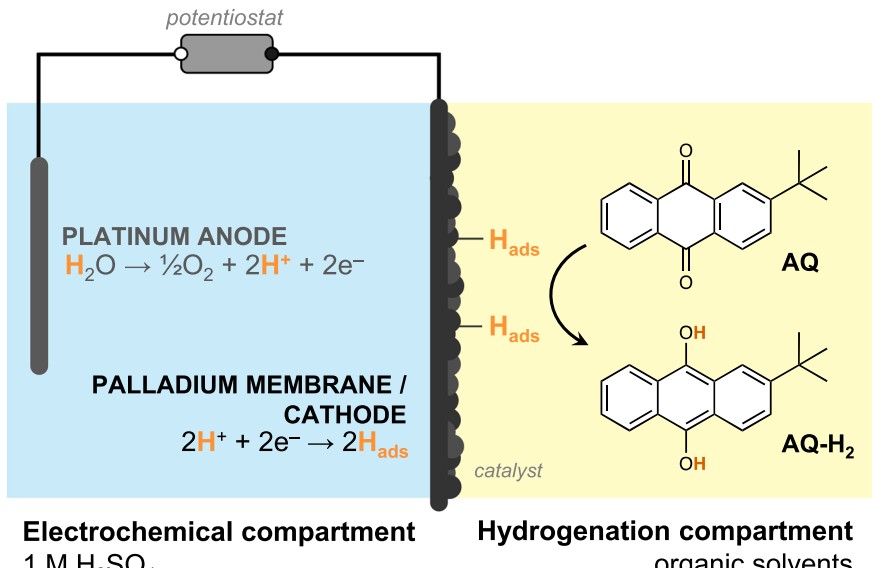

**Fig. 2 | Anthraquinone hydrogenation in a membrane reactor.** Schematic of the reactions taking place in a membrane reactor. Water oxidation occurs at a Pt anode in an aqueous electrolyte (blue) which generates H⁺. Then, the Pd membrane serves as (i) a cathode to reduce H⁺ into H atoms, (ii) a membrane in which H atoms diffuse through interstitial sites, and (iii) a catalyst for hydrogenation of 2-*tert*-butylanthraquinone (AQ) to the hydrogenated product (AQ-H₂) dissolved in an organic solution (yellow).

for H₂O₂ production are provided in Supplementary Fig. 1 and Supplementary Table 1.

Here, we demonstrate that a Pd membrane flow reactor enables the indirect electrochemically-driven synthesis of H₂O₂ at high current densities (100 mA cm⁻²) with high current efficiency (70%) without an H₂ gas supply. These metrics are made possible by using a Pd membrane that physically separates an electrochemical compartment (containing an aqueous electrolyte) and a hydrogenation compartment (containing an anthraquinone dissolved in organic solvents; Fig. 2)[21–26]. In the electrochemical compartment, protons produced from water oxidation at a Pt anode are reduced to monoatomic hydrogen atoms at the surface of a Pd foil that acts as a cathode, a hydrogen-permeable membrane, and a catalyst for hydrogenation. These electrolytically-produced hydrogen atoms subsequently absorb into the interstitial holes of the face-centered cubic (*fcc*) lattice of Pd and permeate through the foil to resurface in the hydrogenation compartment to

react with anthraquinone at a high-surface-area catalyst layer on the Pd foil. This study presents a fast rate of electrochemically-driven anthraquinone hydrogenation (1.32 ± 0.14 mmol h⁻¹ normalized per 1 cm² of geometric surface of electrode). This work offers an electrochemically-driven indirect method for H₂O₂ production.

## Results and Discussion

To demonstrate H₂O₂ synthesis using the membrane reactor, we hydrogenated 0.25 M 2-*tert*-butylanthraquinone (AQ) to form 2-*tert*-butylanthrahydroquinone (AQ-H₂) with 100% conversion and 80 ± 7% current efficiency at 75 mA cm⁻². These results were made possible by delivering AQ in a flow reactor operating at high current densities, and using a solvent composition containing a high amount of aliphatic alcohol (2,6-dimethyl-4-heptanol; DIBC) at elevated temperatures. The AQ-H₂ produced in this reactor can then be mixed with oxygen gas in a separate oxidation reactor to form H₂O₂. We demonstrate stable

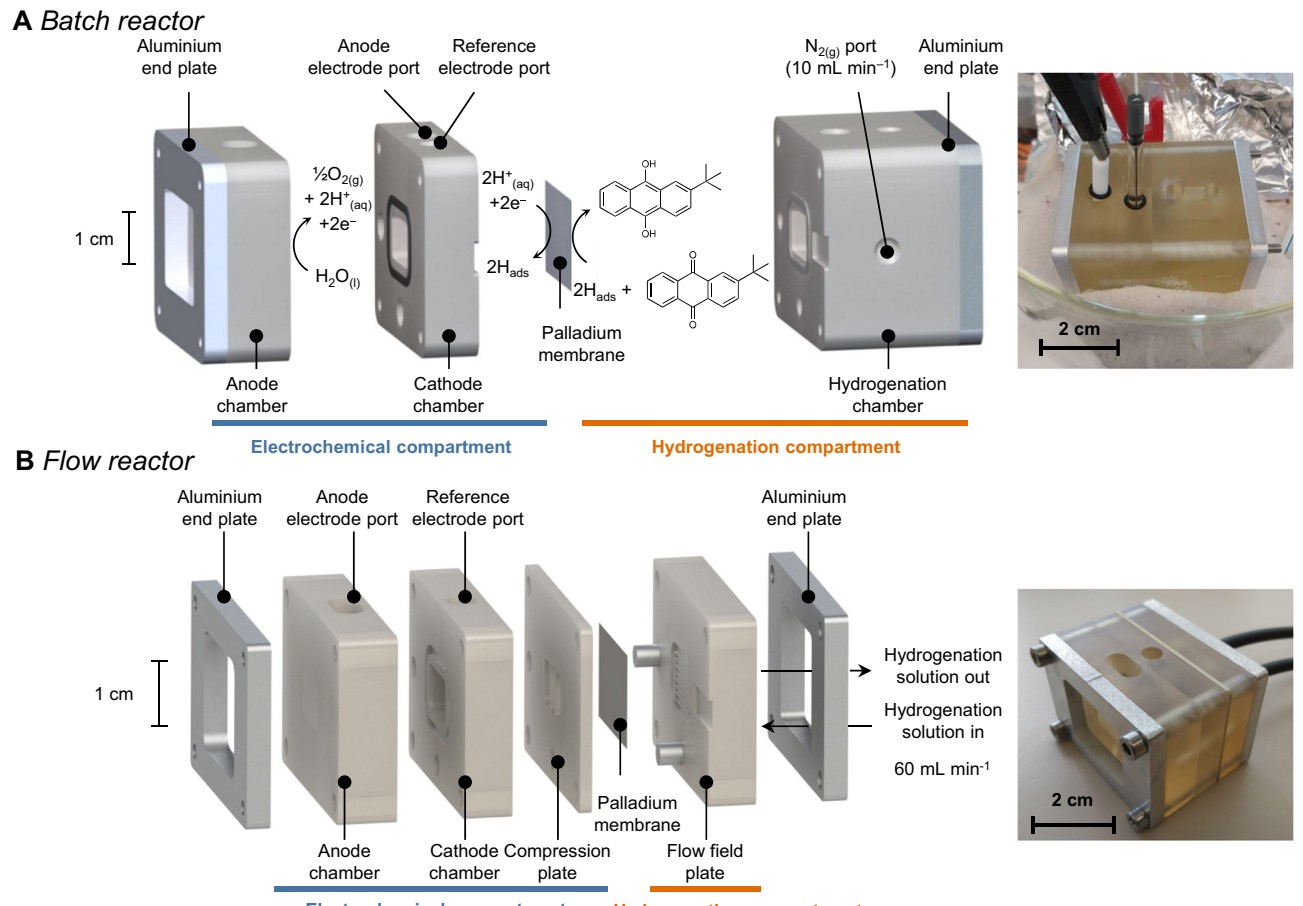

**Fig. 3 | Exploded view of the batch and flow membrane reactors used in this study. A** The batch reactor contains an electrochemical compartment (consisting of an anode chamber and a cathode chamber) and a hydrogenation compartment. The hydrogenation compartment was supplied with $N_{2(g)}$ delivered at 10 mL min$^{-1}$. **B** The flow reactor contains an electrochemical compartment (consisting of an anode compartment, a cathode compartment, and a compression plate) and a hydrogenation compartment (consisting of a flow field plate). The flow field plate was supplied with hydrogenation solution delivered at 60 mL min$^{-1}$. For both reactor architectures, the electrochemical and hydrogenation compartments are separated by a Pd membrane covered with an electrodeposited Pd catalyst facing the hydrogenation compartment. Inset: photographs of the batch (top) and flow (bottom) reactors.

synthesis of $H_2O_2$ over 48 h by continuously recirculating the AQ solution between the membrane reactor and an oxidation reactor.

We performed AQ hydrogenation in two membrane reactor architectures: either (i) a batch reactor (Fig. 3A, Supplementary Figs. 2–4); or (ii) a flow reactor (Fig. 3B, Supplementary Figs. 5–7). Both reactors were made in-house from a high-temperature-resistant resin (Formlabs proprietary resin) and consisted of an electrochemical compartment (8 mL of 1 M $H_2SO_4$) and a hydrogenation compartment separated by a Pd foil. The hydrogenation compartment contained AQ dissolved in a mixture of xylenes (mixture of isomers) and DIBC. The hydrogenation compartment of the batch reactor contained 8 mL of the 0.25 M AQ solution which was magnetically stirred. The hydrogenation compartment of the flow reactor consisted of a flow field plate where 15 mL of the 0.25 M AQ solution was recirculated from a hydrogenation reservoir at 60 mL min$^{-1}$. To perform AQ hydrogenation, a Pt counter electrode and a Ag/AgCl reference electrode were fitted to the electrochemical compartment. A constant current was applied between the Pd foil working electrode and the Pt counter electrode. Aliquots of the AQ solution were periodically sampled to determine the conversion of AQ into AQ-$H_2$ by iodometric titration.

### Proof-of-concept hydrogenation of anthraquinone in a membrane reactor

For the first stage of this study, we set out to demonstrate that an anthraquninone could be hydrogenated in the membrane reactor. We

chose AQ because this molecule is highly soluble (> 0.25 M) at room temperature in xylenes and is a common anthraquinone used in industry. The industry uses 2-alkyl anthraquinones where the alkyl group is typically ethyl, amyl, or *tert*-butyl[3,9]. We ran a proof-of-concept hydrogenation experiment in a batch reactor (Fig. 3A, Supplementary Figs. 2–4) where the electrochemical compartment was filled with 1 M $H_2SO_4$ and the hydrogenation compartment was filled with 0.25 M AQ dissolved in a 1:1 $v/v$ mixture of xylenes:DIBC. DIBC was selected because it can dissolve the slightly polar hydrogenated product (AQ-$H_2$). We ran this experiment for 4 h at a current density of 30 mA cm$^{-2}$ at 50 °C. We then quantified the hydrogenated products by iodometric titration and observed 100% conversion of AQ after 3 h of reaction (Supplementary Fig. 8). While this proof-of-concept study demonstrated that we could indeed hydrogenate AQ, the batch reactor architecture and low applied current density (30 mA cm$^{-2}$) are not applicable for industrial use.

### Membrane reactor optimization

We then studied the effect of continuously flowing the AQ solution into the hydrogenation compartment in a flow reactor. This flow reactor architecture was developed so that it could be combined with a separate oxidation reactor, where $H_2O_2$ is produced by reacting AQ-$H_2$ with $O_2$ gas (Figs. 1B, 3B, and Supplementary Figs. 5–7). Experiments were performed using the flow reactor at a current density of 30 mA cm$^{-2}$ at 50 °C for 2.5 h. We observed that the initial rates of

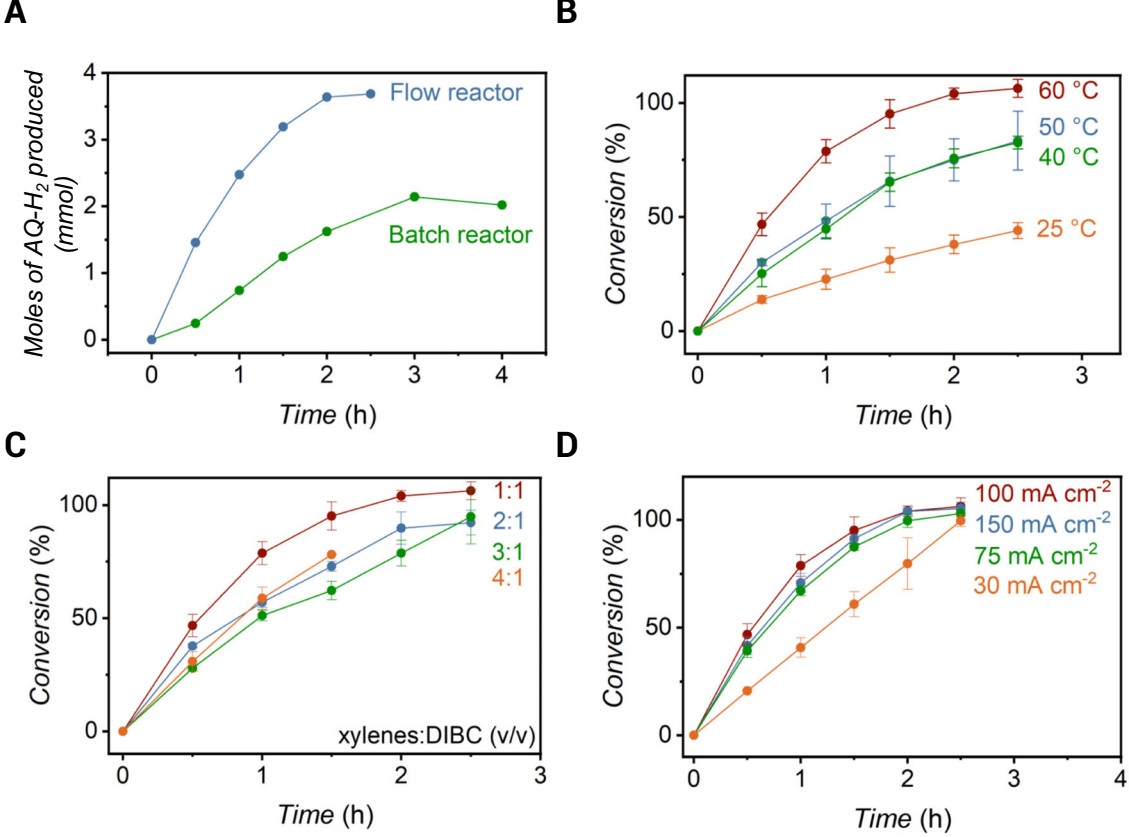

**Fig. 4 | Hydrogenation rates of 2-*tert*-butylanthraquinone (AQ) in a flow reactor. A** Moles of 2-*tert*-butylanthrahydroquinone (AQ-H$_2$) generated as a function of time during electrolysis of AQ in 1:1 *v/v* xylenes:2,6-dimethyl-4-heptanol (DIBC) at 30 mA cm$^{-2}$ and at 50 °C using a batch (green) or a flow (blue) reactor. **B** Conversion of AQ in 1:1 *v/v* xylenes:DIBC as a function of time for various temperatures during electrolysis at 100 mA cm$^{-2}$. **C** Conversion of AQ as a function of time for various solvent compositions (xylenes:DIBC *v/v*) during electrolysis at 100 mA cm$^{-2}$ and at 60 °C. **D** Conversion of AQ in 1:1 *v/v* xylenes:DIBC as a function of time for various current densities at 60 °C. Error bars denote experiments run in triplicate.

hydrogenation in the flow reactor were > 3-fold faster than in the batch reactor (1.02 ± 0.16 mmol h$^{-1}$ *c.f.* 0.28 ± 0.07 mmol h$^{-1}$ per 1 cm$^2$ of the geometric area of Pd foil, respectively; Fig. 4A). Initial hydrogenation rates were determined by the slope of AQ conversion versus time over the first 30 min of electrolysis, and are referred to herein as hydrogenation rates. The increase in hydrogenation rate within the flow reactor is consistent with our previously reported work, where we measured an initial 2-fold higher hydrogenation rate of phenylacetylene in a flow reactor compared to a batch reactor[26]. This increase is attributed to higher mass transport in the flow reactor. We performed in-situ atmospheric−mass spectrometry (atm−MS) in the batch and flow reactors to quantify the hydrogen permeating through to the hydrogenation compartment, and the H$_{2(g)}$ that evolves in the electrochemical compartment. These experiments were performed without AQ initially in the hydrogenation solution, at 75 mA cm$^{-2}$, without N$_2$ gas, and at 50 °C. We observed 43% (0.334 μmol$_H$ s$^{-1}$ cm$^{-2}$) and 99% (0.770 μmol$_H$ s$^{-1}$ cm$^{-2}$) H permeating to the hydrogenation compartment for the batch and flow reactors, respectively (Supplementary Figs. 9, 10). (The amount of hydrogen permeating to the hydrogen compartment was calculated by measuring the amount of hydrogen evolved in the hydrogenation compartment divided by the total amount of hydrogen produced in the hydrogenation and electrochemical compartments.) When we added AQ to the hydrogenation compartment (to reach 0.25 M AQ in the reaction vessel), we observed 73% and 64% decreases in H$_2$ evolution in the hydrogenation compartment of the batch and flow reactors, respectively. We used the decrease in the amount of H$_2$ evolved in the hydrogenation compartment as a measure of the amount of H atoms that reacted with AQ. The

current efficiencies calculated based on atm−MS experiments (30% for the batch reactor and 64% for the flow cell) were in good agreement with the current efficiencies calculated by independent titration experiments (27% for the batch reactor after 30 min of electrolysis, and 69% for the flow cell after 1 h of electrolysis for experiments at 75 mA cm$^{-2}$). We attribute the increase in current efficiency to the increase in mass transport rate, which is a result of the continuous supply of AQ to the catalyst surface available to react with the hydrogen produced in the flow reactor, a behavior that is also observed in hydrogen fuel cells and CO$_2$ electrolyzers[27,28].

The next goal of this study was to modify temperature, solvent composition, and current density to ensure that anthraquinone hydrogenation in the membrane reactor was competitive with industrial standards. Industrial anthraquinone hydrogenation is generally performed at moderate temperatures (40–70 °C) with a Pd catalyst to ensure solubility of the reactants and products[29–31]. We tested the effect of various temperatures (25, 40, 50, 60 °C) on the hydrogenation rates during electrolysis at 100 mA cm$^{-2}$ and in 1:1 *v/v* xylenes:DIBC. We observed the fastest hydrogenation rates at the highest temperature tested (60 °C, Fig. 4B). The hydrogenation rates were found to be similar for intermediate temperatures (40 and 50 °C), and significantly slower at room temperature (25 °C). We did not observe product precipitation at any of the temperatures we tested in the reaction vessel due to the dark color of the solution, but we did observe precipitates when sampling for analysis. A decrease in the maximum AQ conversion was also observed as temperature was lowered (i.e., 106.3 ± 3.4%, 83.4 ± 12.9%, 82.6 ± 2.8%, and 44.0 ± 3.5% at 60, 50, 40, and 25 °C, respectively)

during the course of the 2.5 h reaction. This dependence of temperature on hydrogenation rates is comparable to the results obtained for the hydrogenation of anthraquinones with $H_2$ gas, where the hydrogenation of anthraquinones is reported to be temperature-dependent[32,33].

Industrial anthraquinone hydrogenation is performed in a mixture of apolar and polar solvents required to solubilize the reactant and product, respectively. We tested the effect of various volume ratios (from 1:1 to 4:1) of xylenes (an apolar solvent) to DIBC (a polar solvent) on the hydrogenation rates during electrolysis at 100 mA cm$^{-2}$ and at 60 °C. We observed faster hydrogenation rates when a 1:1 $v/v$ xylenes:DIBC mixture was used. However, comparable hydrogenation rates were achieved for all other solvent compositions (Fig. 4C). We observed product precipitation after 1.5 h of reaction when a 4:1 $v/v$ xylenes:DIBC solution was employed, which led us to stop the experiment. These results suggest that the formation of AQ·$H_2$ is promoted by high fractions of DIBC because of the higher product solubility.

Finally, we tested various current densities because this parameter is directly related to hydrogenation rates[21,22]. We observed that the hydrogenation rates indeed increased when the current density was increased from 30 mA cm$^{-2}$ to 100 mA cm$^{-2}$, but that the hydrogenation rates at 150 mA cm$^{-2}$ were within experimental error of those achieved at 100 mA cm$^{-2}$ (Fig. 4D). We analyzed H permeation through the Pd membrane by atm−MS and observed that hydrogen permeation increased when the current density was increased from 30 to 150 mA cm$^{-2}$ (Supplementary Fig. 11). We therefore interpret that the hydrogenation rates were not limited by hydrogen availability at the hydrogenation catalyst surface. These results are promising because they suggest that the membrane reactor can be scaled to current densities relevant for industry. Although the hydrogenation rates increased with current density, the current efficiencies (see *Methods, Current efficiency* section for details) decreased linearly with current density. We observed current efficiencies of $105 \pm 2\%$, $80 \pm 7\%$, $71 \pm 8\%$, and $42 \pm 4\%$ during electrolysis at 30, 75, 100, and 150 mA cm$^{-2}$, respectively.

## Demonstration of $H_2O_2$ synthesis from an anthraquinone hydrogenated in a membrane reactor

We have shown that a flow membrane reactor enables fast hydrogenation rates of AQ, however, this architecture is only useful if it can be used to produce $H_2O_2$ in a closed loop. We next integrated an oxidation reactor into our process such that the outlet hydrogenation solution from the membrane reactor was pumped directly into the oxidation reactor (Supplementary Figs. 12, 13). The oxidation reactor contained an organic AQ solution and a denser, immiscible aqueous solution (Fig. 5A). Air was bubbled at 90 mL min$^{-1}$ into the aqueous solution. AQ·$H_2$ in the hydrogenation solution reacted with $O_2$ gas to produce $H_2O_2$ which diffused from the organic phase to the aqueous phase, and AQ which remained in the organic phase. The AQ solution was then gravity-fed to a reservoir for heating before being fed back to the membrane reactor for hydrogenation.

We performed AQ hydrogenation in the flow reactor at 75 mA cm$^{-2}$ while continuously recirculating the AQ solution between the flow reactor and the oxidation reactor for 48 h. We regularly quantified $[H_2O_2]$ in the aqueous phase of the oxidation reactor by

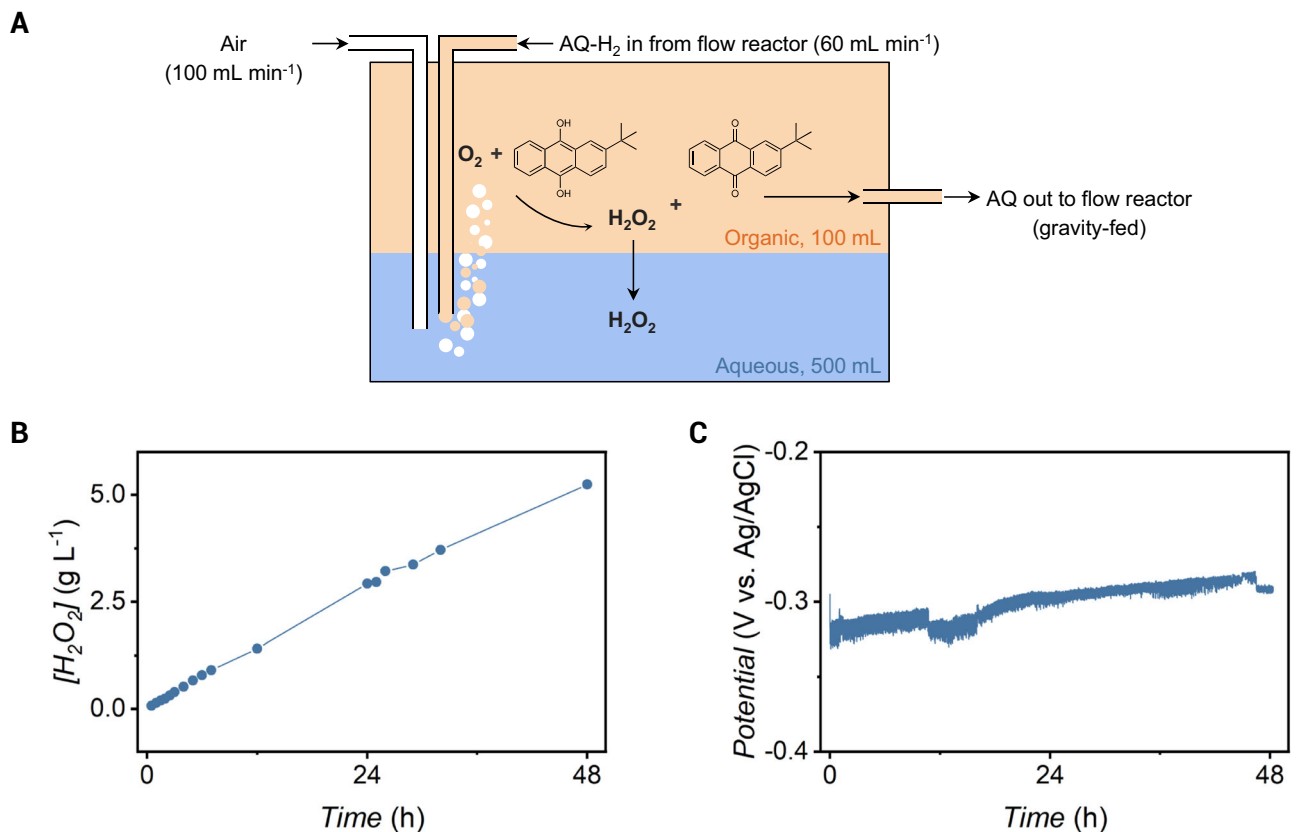

**Fig. 5 | Integration of 2-*tert*-butylanthraquinone (AQ) hydrogenation in the flow reactor with $H_2O_2$ synthesis in an oxidation reactor in a recirculated system. A** Schematic of the oxidation reactor containing an aqueous phase (500 mL) immiscible to the hydrogenation organic solution (100 mL). 2-*tert*-Buty-lanthrahydroquinone (AQ·$H_2$) is fed from the flow reactor and reacts with $O_2$ contained in the air fed at 100 mL min$^{-1}$ to form $H_2O_2$ and AQ. $H_2O_2$ then transfers to the aqueous phase while AQ solution is gravity fed back to the flow reactor for further hydrogenation. **B** $[H_2O_2]$ in the aqueous layer of the oxidation reactor as measured by permanganate titrations as a function of time during electrolysis at 75 mA cm$^{-2}$. **C** Cathode potential as a function of time during electrolysis at 75 mA cm$^{-2}$.

permanganate titrations. We observed a stable synthesis of $H_2O_2$ over 48 h of electrolysis ($[H_2O_2]$ = 5.24 g L$^{-1}$ at 48 h) with a current efficiency of 54 ± 4% (Fig. 5B). We also recorded the cathodic potential in the electrochemical compartment and observed that this potential remained stable at −0.30 ± 0.01 V $vs.$ Ag/AgCl (Fig. 5C). Collectively, these results show that a membrane reactor can be used to produce $H_2O_2$ and that this reactor is stable over 48 h.

The results presented in this study highlight the power of separating water oxidation from hydrogenation in a membrane reactor. This reactor enables the electrochemically-driven hydrogenation of an anthraquinone in organic media that subsequently generate $H_2O_2$. This process uses renewable electricity and water instead of $H_2$ gas. The two-compartment reactor architecture enabled a 3-fold improvement in hydrogenation rates and 6-fold higher current densities compared to previous reports of electrochemical hydrogenation of anthraquinones. The use of a membrane reactor to perform hydrogenation chemistry at moderate temperatures and ambient pressures from electrolytically-sourced hydrogen is a significant step forward to decarbonize industrial $H_2O_2$ production.

## Methods

### Materials
Pd (99.95%) was purchased as a 1-oz wafer bar from Silver Gold Bull. Hydrogen peroxide ($H_2O_2$, 30% (w/w) in $H_2O$, contains stabilizer), xylenes (mixture of isomers, > 98.5% + ethylbenzene basis), 2,6-dimethyl-4-heptanol (DIBC, 80%), 2-$tert$-butylanthraquinone (AQ, 98%), iodine ($I_2$, > 99.8%), potassium iodide (KI, 99%), potassium permanganate ($KMnO_4$, > 99.0%), sodium thiosulfate ($Na_2S_2O_3$, > 99.99%), tetrabutylammonium hexafluorophosphate (TBA-PF$_6$, > 99%), dichloromethane ($CH_2Cl_2$, > 99.8%), palladium chloride ($PdCl_2$, > 99.9%), acetonitrile ($CH_3CN$, anhydrous, 99.8%), sulfuric acid ($H_2SO_4$, 95.0–98.0%), hydrochloric acid (HCl, 37%), nitric acid ($HNO_3$, 70%), ethylenediaminetetraacetic acid (EDTA, 99%), citric acid, (≥ 99.5%), and sodium acetate (NaOAc, > 99.0%) were purchased from Sigma Aldrich. Acetic acid (glacial, HOAc) and starch indicator (1% (w/v) aqueous solution, Ricca Chemical) were purchased from Fischer Scientific. Pt gauze (52 mesh, 99.9%) was purchased from Alfa Aesar. All reagents were used without further purification unless specifically mentioned. Ag/AgCl reference electrodes (RE5B) were purchased from BASi. Pt counter electrodes (CHI115) were purchased from CH Instruments. High-temperature-resistant proprietary resin was purchased from Formlabs. Stainless steel dowel pins, stainless steel sheets, stainless steel fasteners, and Viton o-rings were purchased from McMaster-Carr.

### Reactor design
**Batch hydrogenation reactor.** The hydrogenation chamber, cathode chamber, and anode chamber (Supplementary Fig. 3) of the batch reactor were printed by a stereolithography 3D printer. The chambers are composed of high-temperature-resistant resin (Formlabs proprietary resin). The Pd membrane was sealed between the components by Viton o-ring gaskets (square cross-section), and the assembly was clamped together by two waterjet cut ¼"-thick aluminum end plates connected by four M4 stainless steel bolts. Quick-turn polycarbonate couplings (¼"-28 UNF thread size) were used to feed $N_2$ gas to the hydrogenation compartment of the batch reactor (the compartment was printed with the corresponding threads). Hydrogenation in a batch reactor was performed at atmospheric pressure. The electrochemical compartment was exposed to air while the hydrogenation compartment was constantly bubbled with $N_2$ at 10 mL min$^{-1}$ to prevent oxidation of the produced 2-$tert$-butylanthrahydroquinone (AQ-$H_2$). The hydrogenation compartment was filled with 8 mL of a solution of AQ (0.25 M) dissolved in a mixture of xylenes and DIBC (1:1 $v/v$). The hydrogenation solution was stirred with a magnetic bar. 8 mL of 1 M $H_2SO_4$ was added to the electrochemical compartment.

The batch reactor was immersed in a sand bath and heated until the temperature of the solution in the electrochemical compartment was stable at 50 °C. A constant current density of 30 mA cm$^{-2}$ was applied for up to 5 h. Aliquots of the hydrogenation solution were sampled every 30 min or 1 h to monitor reaction progress by iodometric titration.

**Flow hydrogenation reactor.** The flow field plate, compression plate, cathode chamber, and anode chamber (Supplementary Fig. 6) of the flow reactor were printed by a stereolithography 3D printer. These components are composed of high-temperature-resistant resin (Formlabs proprietary resin). The Pd membrane was sealed between the compression plate and the flow field plate by Viton o-ring gaskets (square cross-section), and the assembly was clamped together by two waterjet cut ¼"-thick aluminum end plates connected by four M4 stainless steel bolts. Quick-turn PVDF plastic couplings (¼"-28 UNF thread size) were used as the inlet and outlet feeds of the AQ solution to the hydrogenation compartment of the flow reactor (the compartment was printed with the corresponding threads). The hydrogenation plate contained a serpentine flow field consisting of seven 1 × 1 mm flow channels and six 0.7 mm-wide ribs. Hydrogenation in a flow reactor was performed at atmospheric pressure. The electrochemical compartment was exposed to air while the hydrogenation reservoir was constantly bubbled with $N_2$ at a flow rate of 10 mL min$^{-1}$. The hydrogenation reservoir was filled with 15 mL of a solution of AQ of various concentrations dissolved in a mixture of xylenes and DIBC of various compositions. The hydrogenation solution was recirculated at 60 mL min$^{-1}$ between the hydrogenation plate and the hydrogenation reservoir by means of a low-flow chemical metering pump (4049K55, McMaster-Carr). 8 mL of 1 M $H_2SO_4$ was added to the electrochemical compartment. The flow reactor and the hydrogenation reservoir were immersed in a sand bath and heated until the temperature of the solution in the electrochemical compartment was stable at the desired temperature (i.e., 25, 40, 50, or 60 °C). A constant current density was applied for up to 3 h and aliquots of the hydrogenation solution were sampled periodically to monitor reaction progress by iodometric titration.

### Pd foil preparation
Pd foils were rolled from a 1-oz Pd wafer bar to ~150 μm with a mechanical roller. The bar was further rolled by means of an electric rolling mill (MTI MR-100A) to 25 μm thick. The thickness of each Pd foils was determined by a Mitutoyo digital micrometer. The Pd foils were annealed at 850 °C for 1.5 h in an $N_2$ atmosphere and subsequently cleaned using a 1:2:1 $HNO_3$:$H_2O_2$:$H_2O$ $v/v$ ratio solution until vigorous bubbling subsided (~20 min). Electrodeposition was performed immediately following this cleaning step (see catalyst preparation below).

### Catalyst preparation
A high-surface-area Pd catalyst was electrodeposited onto the foil to increase the hydrogenation reaction rate (Supplementary Fig. 14). The electrochemical compartment was filled with 8 mL of 15.9 mM $PdCl_2$ in 1 M HCl and a Ag/AgCl reference electrode and Pt mesh counter electrode were fitted to the cell. −0.2 V $vs.$ Ag/AgCl was applied to the working electrode foil until 30 C of charge had been passed. This deposition procedure was done twice for a total of 60 C of charge passed (~10 mg of electrodeposited Pd, the rest of the charge accounts for $H_2$ evolution). The deposition current was approximately 20–30 mA. The resulting black-colored Pd catalyst was used for a maximum of 10 reactions before being removed from the foil using $HNO_3$ and gentle mechanical abrasion. The foil was subsequently cleaned and replated with a new catalyst for continued use. The Pd foils were reused until pinholes formed in the foil as a result of: (i) hydrogen embrittlement from repeated hydrogen

absorption and desorption; (ii) etching of the foil during cleaning; and/or (iii) repeated clamping of the foil in the flow reactor[26]. The Pd membranes were reused for > 20 reactions before pinhole formation occurred.

## Electrochemical active surface area measurements

The electrochemical active surface area (ECSA) measurement of the electrodeposited Pd catalyst was performed in the batch cell with catalyst-coated surface exposed to the electrochemical compartment. The electrochemical compartment was filled with 8 mL of 0.15 M TBA-PF$_6$ dissolved in dry acetonitrile and a Ag/AgCl reference electrode and a Pt mesh (1 cm$^2$) were fitted to the cell. The open-circuit potential (OCP) was measured against Ag/AgCl and 3-cycles of cyclic voltammograms were performed from the OCP to ± 50 mV vs. OCP at various scan rates (20–50 mV s$^{-1}$) (Supplementary Fig. 15). The difference between anodic and cathodic currents at the OCP on the 3rd cycle was plotted against the scan rate (Supplementary Fig. 16). We found a real capacitance of 3.833 mF cm$^{-2}$ by extrapolating the slope of the plot. Surendranath and coworkers reported a specific capacitance for Pd to be approximately 11 μF cm$^{-2}$ [34]. Therefore, we found an ECSA of 857 cm$^2$ which is associated with a 348-fold increase in surface area compared to a "bare" Pd foil.

## Electrochemistry

A Metrohm Autolab PGSTAT302N potentiostat was used to control the electrochemical experiments. A Pd foil was fitted between the electrochemical compartment and either the hydrogenation flow field (for experiments with the flow reactor) or the hydrogenation compartment (for experiments with the batch reactor). Viton o-rings were used to seal both the components of the cell to prevent the solutions from leaking. The electrolysis compartment was filled with 8 mL of 1 M H$_2$SO$_4$, and a Ag/AgCl reference electrode (saturated with KCl) and a Pt mesh counter electrode were fitted to the cell. Electrolysis was driven galvanostatically, where a reductive current was applied by the potentiostat to the Pd foil working electrode. The potential ($E_{Ag/AgCl}$) was measured between the Pd foil cathode and Pt mesh anode. No iR correction was performed because the high salinity of the electrolyte and close proximity of the reference electrode to the Pd foil cathode (~3 mm) resulted in negligibly small uncompensated resistances of < 0.1 Ω. The thickness of the foil was 25 μm and the geometric surface area of the foil was 2.46 cm$^2$. All experiments in the flow reactor were performed in triplicate (except for Fig. 4A) with fresh hydrogenation and electrochemical solutions.

## Iodometric titrations

The conversion of AQ to AQ·H$_2$ was quantified by iodometric titration adapted from a previous study[35]. In short, 2 mL of an acetate buffer solution (0.1 M H$_2$SO$_4$, 0.5 M NaOAc, and 0.05 M HOAc), 0.65 mL of iodine solution (50 mM I$_2$ and 250 mM KI), and 0.1 mL of the hydrogenation solution containing the AQ·H$_2$ were stirred under 10 mL min$^{-1}$ of N$_2$ for 5 min for the following reaction to be complete:

$$AQ - H_2 + I_2 \rightarrow AQ + 2HI \qquad (5)$$

Then, any I$_2$ remaining in excess (I$_{2,excess}$) was titrated against a 2 mM thiosulfate solution until the purple color of the starch indicator disappeared (Eq. 6, Supplementary Fig. 17).

$$I_{2,excess} + 2S_2O_3^{2-} \rightarrow 2I^- + S_4O_6^{2-} \qquad (6)$$

The volume used to titrate I$_{2,excess}$ ($V_{excess}$) was subtracted to the volume necessary to titrate the total I$_2$ initially in solution without AQ-H$_2$ ($V_O$, this titration was performed in triplicate for each experiment) which correlates to the amount of I$_2$ that reacted with AQ-H$_2$ ($V_{reacted}$).

The conversion of AQ to AQ·H$_2$ was determined with the following equation:

$$Conversion\,[\%] = \frac{n_{AQ-H2}\,[mol]}{n_{AQ,initial}\,[mol]} = \frac{V_{reacted}\,[mL] \times C_{S2O32-}\,[M]}{2 \times V_{AQ,initial}\,[mL] \times C_{AQ,initial}\,[M]} \times 100 \qquad (7)$$

Where $n_{AQ·H2}$ is the number of moles of AQ·H$_2$ produced, $n_{AQ, initial}$ is the number of moles initially in solution before reaction, $V_{reacted}$ is defined in the sentence above, $C_{S2O32-}$ is the thiosulfate concentration (2 mM), $V_{AQ,initial}$ is the volume of hydrogenation solution added to the titration (0.1 mL), and $C_{AQ,initial}$ is the concentration of AQ before reaction (0.25 M).

## Current efficiency

We define the current efficiency (CE) of the reaction as the percentage of electrons supplied to the Pd foil that were used to indirectly hydrogenate AQ (through proton reduction in the cathode chamber followed by H permeation through the Pd foil into the hydrogenation chamber). CE was calculated with following equation:

$$CE\,[\%] = \frac{2n_{AQ-H2}}{n_{e-}} = \frac{2 \times Conversion\,[\%] \times C_{AQ,initial}\,[M] \times V_{reaction}\,[L] \times F\,[C\,mol^{-1}]}{J\,[A\,cm^{-2}] \times A\,[cm^2] \times t\,[s]} \times 100 \qquad (8)$$

Where $n_{AQ·H2}$ is the number of moles of AQ·H$_2$ produced, $n_{e-}$ is the moles of electrons supplied to the cathode, Conversion is the percentage of AQ that has been hydrogenated (determined by titration), $C_{AQ,initial}$ is the initial concentration of AQ, $V_{reaction}$ is the total volume of hydrogenation solution (0.015 L), $F$ is Faraday's constant (96,485 C mol$^{-1}$), $J$ is the current density applied to the cell, $A$ is the geometric surface area of the catalyst (2.46 cm$^2$), and $t$ is the time of the reaction. This calculation accounts for losses due to H$_{2(g)}$ evolution both in the electrochemical and hydrogenation compartments.

## Gas chromatography–mass spectrometry (GC–MS)

GC–MS was performed to ensure that no undesired side-product (see Supplementary Figs. 18, 19) was produced. 0.1 mL of the hydrogenation reaction at t = 0 and after 3 h of reactions were sampled and diluted in 1 mL of dichloromethane. GC–MS experiments were conducted on an Agilent GC–MS using an HP-5 ms column and electron ionization. The samples were run using an auto-sampler with a 1-L injection volume and a split ratio of 20:1. The oven temperature was static at 50 °C for 1 min, then the temperature was ramped to 280 °C at a 25 °C min$^{-1}$ rate, and finally, the temperature was held at 280 °C for 5 min. A solvent delay of 5.1 min was employed. The peak for AQ was identified at 11.2 min by searching the NIST database for matching mass spectra (Supplementary Fig. 19). No other peaks were found in the GC spectra.

## Atmospheric–mass spectrometry (atm–MS)

The batch and flow reactors for the atm–MS experiments were set up the same as for the hydrogenation experiments (described in the Reactor design section) with the following modifications to enable monitoring of hydrogen evolution in the hydrogenation compartment: (i) the hydrogenation solution initially contained no AQ; (ii) we did not flow N$_2$ in the hydrogenation reservoir; and (iii) a condenser was used between the hydrogenation reservoir and the atmospheric–mass spectrometer to ensure no vapors of solvents reached the spectrometer detector. The 2 m/z ion current was continuously monitored using an ESS CatalySys atm–MS with a flow rate into the instrument of 10 mL min$^{-1}$. A constant current density was applied between the Pt anode and the Pd foil cathode until H$_2$ evolution had equilibrated, then the current density was increased to 75 mA cm$^{-2}$ where it was held for the duration of the experiment (Supplementary Figs. 9, 10).

Data was recorded using a single channel until a steady signal was observed for 30 min in each compartment prior to AQ addition. The spectrometer ran and collected data continuously during the inlet changes between compartments. AQ was then added to the hydrogenation compartment until a 0.25 M concentration was reached and the hydrogen evolution was monitored in the compartment until a steady signal was recorded. Results were plotted over time only to show the stability of the signal. The inlet change during the experiment occurs on the order of seconds, while the data collected was on the order of hours. Hydrogen permeation was calculated with the following equation:

$$H_{permeation} \, [\%] = \frac{H_{2,hydrogenation}^{+}}{(H_{2,hydrogenation}^{+} + H_{2,electrochemical}^{+})} \times 100 \qquad (9)$$

Where $H_{2,hydrogenation}^{+}$ is the stabilized ion current recorded in the hydrogenation chamber and $H_{2,electrochemical}^{+}$ is the stable ion current recorded in the electrochemical chamber. The ion current is proportional to $H_2$ permeation through the Pd membrane.

## Data availability

The source data supporting the figures is available in the source file. Additional data supporting the findings in this study are available either within the paper or its Supplementary Information, or from the corresponding author on request. Source data are provided with this paper.

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

## Acknowledgements

We thank Benjamin Herring at the UBC Shared Instrument Facility for assistance with GC–MS measurements.

## Author contributions

A.G.F. and R.S.D. conceived the study. A.G.F. and C.H. designed the experiments. A.G.F. performed the experiments and generated the figures. A.R.R. performed some of the atm–MS experiments. N.E.L. assisted with atm–MS experiments. A.H. helped with titration experiments. A.G.F. and C.H. wrote the first manuscript draft. M.S. completed revisions of the manuscript. C.P.B. supervised the project and conceived the study. All authors contributed to the manuscript writing.

## Funding

We are grateful to Solvay SA, the Stewart Blusson Quantum Matter Institute's Quantum Electronic Science and Technology Program, Canadian Natural Sciences and Research Engineering Council (RGPIN-2018-06748), Canadian Foundation for Innovation (229288), Canadian Institute for Advanced Research (BSE-BERL-162173), Canada Research Chairs, and the Canada First Research Excellence Fund, Quantum Materials and Future Technologies Program for financial support. Arthur G. Fink acknowledges funding from the UBC Four Year Doctoral Fellowship program.

## Competing interests

(1) A U.S. patent based on the technology described in this work has been granted, and a patent application has been filed with the European Patent Office: Berlinguette, C. P.; Huang, A.; Delima, R. S.; Jansonius, R. P. "Methods and Apparatus for Producing Hydrogen Peroxide." US Patent No. US11761105B2, granted September 2023. European Patent Application No. 22783753.1, filed November 2023. Priority data: US Provisional Patent Application No. 63/173,138 and No. 63/173,745, filed April 2021. (2) A PCT application based on the technology described in this work has been filed: Berlinguette, C. P.; Fink, A. G.; Schwiedernoch, R.; Diaz-Maroto Carpintero, J. "Methods And Apparatus For Indirect Production Of Hydrogen Peroxide Using Amyl-Anthraquinone For Hydrogen Transport." PCT Application No. PCT/CN2022/119784, filed September 2022. (3) Additional patent applications based on the palladium membrane reactor technology described in this work have been filed (one has been granted): Berlinguette, C. P.; Sherbo, R. S. "Methods and Apparatus for Performing Chemical and Electrochemical Reactions". Canadian Patent Application No. 3089508, Publication No. CA3089508 (published August 2019), pending. US Patent No. US11667592B2, granted June 2023. US Patent Application No. 18/305,970, Publication No. US20230257325A1 (published August 2023), pending. Priority data: US Provisional Patent Application No. 62/622,305, filed January 2018. (4) The remaining authors declare no other competing interests.
