## [Peer Review File · Nature Communications]

REVIEWER COMMENTS

Reviewer #4 (Remarks to the Author):

I was asked whether I could review authors revision to reviewer 2 and 3. As I was reading the article I was in most part agreed with reviewers comment. Which means that reviewers comments were poorly answered.

My specific comments are.

Reviewer 2 Comment 1

Supp Fig 7 & 8 are difficult to understand. As far as I understand first two hours were measured at the electrochemical compartment and some how you switched your H₂ detection at around 2 hr... But by drawing x-axis as time. It was difficult for me to understand this experiment, and I am unsure whether I understood the data correctly.

Reviewer 2 Comment 2

It looks like authors did not get the point of the question. In your work, the cathode catalyst is Pd, and it should catalyze hydrogenation reaction of AQ. So the high production rate is mostly due to the use of Pd. However, you have compared your work with some of AQ electrocatalysts papers. And it seems unfair to compare your work with theirs where the other works did not use Pd catalyst.

Reviewer 2 Comment 3

Authors did not properly argue why flow system enhances efficiency of the system. Is it because it changes diffusion of the species?

Reviewer 3 Comment 1

The work from Haotian Wang also demonstrated water oxidation as a proton source. So supplementary figure 1 should be redrawn, your manuscript should be revised, and also you should compare your work with theirs.

Reviewer 3 Comment 2

Supplementary figure 1 should be revised and you should do more literature study on the matter of electrochemical H₂O₂ formation. For example, your drawing of (b) is misleading

black (reviewers' comments) | blue (authors' response) | **bold** (manuscript changes)

REVIEWER COMMENTS

Reviewer #4 (Remarks to the Author):

I was asked whether I could review authors revision to reviewer 2 and 3. As I was reading the article I was in most part agreed with reviewers comment. Which means that reviewers comments were poorly answered.

My specific comments are.

Reviewer 2 Comment 1

Supp Fig 7 & 8 are difficult to understand. As far as I understand first two hours were measured at the electrochemical compartment and some how you switched your H₂ detection at around 2 hr... But by drawing x-axis as time. It was difficult for me to understand this experiment, and I am unsure whether I understood the data correctly.

We added an explanation in the "*Methods -> Atmospheric-mass spectrometry (atm-MS)*" section of the main text (page 21 of the tracked document; page 21 of the final document) about our data collection and an equation for calculating hydrogen permeation (eq. 9). We have updated SI Fig 7 and 8 to provide clarity for the reviewer and readers.

"A constant current density was applied between the Pt anode and the Pd foil cathode until H₂ evolution had equilibrated, then the current density was increased to **75 mA cm⁻² where it was held for the duration of the experiment (Supplementary Fig. 11). Data was recorded using a single atm-MS channel until a steady signal is observed for 30 min in each chamber prior to AQ addition. The hydrogenation compartment was then added AQ until a 0.25 M concentration was reached and the hydrogen evolution was monitored in the compartment until a steady signal was recorded. Results were plotted over time only to show the stability of the signal.**

$$H_{permeation} = \frac{H_{2,hydrogenation}^+}{(H_{2,hydrogenation}^+ + H_{2,electrochemical}^+)} \times 100\% \quad (9)$$

Reviewer 2 Comment 2

It looks like authors did not get the point of the question. In your work, the cathode catalyst is Pd, and it should catalyze hydrogenation reaction of AQ. So the high production rate is mostly due to the use of Pd. However, you have compared your work with some of AQ electrocatalysts papers. And it seems unfair to compare your work with theirs where the other works did not use Pd catalyst.

We agree that Pd is a catalyst. However, I still contend that our system is electrochemical (the hydrogen comes from water electrolysis).

While we stand by this claim, it is creating a distracting debate. We therefore removed our claim in the introduction that our system is the *fastest*.

Reviewer 2 Comment 3

Authors did not properly argue why flow system enhances efficiency of the system. Is it because it changes diffusion of the species?

This is due to higher mass transport in the flow reactor.

We added text and citations clarifying this claim in the main manuscript section "*Membrane reactor optimization for industrial relevance*" (see page 9 of the tracked document; page 9 of the final document).

“The increase in hydrogenation rates within the flow reactor is consistent with our previously reported work, where we measured an initial 2-fold higher hydrogenation rate of phenylacetylene in a flow reactor compared to a batch reactor²⁷. This increase is attributed to higher mass transport in the flow reactor. We performed *in-situ* atmospheric mass spectrometry (atm-MS) in the batch and flow reactors to quantify the hydrogen permeating through to the hydrogenation compartment, and the H_{2(g)} that evolves in the electrochemical compartments. These experiments were performed without *tert*-butyl anthraquinone initially in the hydrogenation solution, at 75 mA cm⁻², without N₂ gas, and at 50 °C. We observed 43% (0.334 μmol_H s⁻¹ cm⁻²) and 99% (0.770 μmol_H s⁻¹ cm⁻²) H permeating to the hydrogen compartment for the batch and flow reactors, respectively (Supplementary Figs. 7 and 8). (The amount of hydrogen permeating to the hydrogen compartment was calculated by measuring the

amount of hydrogen evolved in the hydrogenation compartment divided by the total amount of hydrogen produced in the hydrogenation and electrochemical compartments.) When we added *tert*-butyl anthraquinone to the hydrogenation compartment (to reach 0.25 M *tert*-butyl anthraquinone in the reaction vessel), we observed 73% and 64% decreases in H₂ evolution in the hydrogenation compartments of the batch and flow reactors. We used the decrease in the amount of H₂ evolved in the hydrogenation compartment as a measure of the amount of H atoms that reacted with anthraquinone. The current efficiencies calculated based on atm-MS experiments (30% for the batch reactor and 64% for the flow cell) were in good agreement with the current efficiencies calculated by independent titrations experiments (27% for the batch reactor after 30 min of electrolysis, and 69% for the flow cell after 1 h of electrolysis for experiments at 75 mA cm⁻²). **We attribute the increase in current efficiency to the increase in mass transport rates by the continuous supply of anthraquinone to the catalyst surface available to react with the hydrogen produced in the flow reactor, behavior that is also observed in hydrogen fuel cells and CO₂ electrolyzers^{28,29}.**”

Reviewer 3 Comment 1

The work from Haotian Wang also demonstrated water oxidation as a proton source. So supplementary figure 1 should be redrawn, your manuscript should be revised, and also you should compare your work with theirs.

Done. The following updates have been made:

- Supplementary Fig 1 has been updated to show several different direct electrochemical approaches to H₂O₂ production

Direct H₂O₂ Synthesis

a) Direct synthesis

b) Direct electrosynthesis

c) Direct synthesis using a Pd membrane reactor

Indirect H₂O₂ Synthesis

d) Industrial process (Riedl-Pfleiderer process)

e) Electrochemical hydrogenation of anthraquinone

f) This work - Indirect synthesis using a Pd membrane reactor

“**Supplementary Fig. 1** | Schematic of hydrogen peroxide (H_2O_2) synthesis by direct and indirect methods; a) Direct synthesis requires that both H_2 and O_2 gases are present in a single reactor at high pressure over a catalyst suspension¹; b) For electrochemical H_2O_2 synthesis; i) water is oxidized at the anode in the presence of an electrolyte and reacts with O_2 at the cathode to generate H_2O_2 ²; ii) H_2 gas is oxidized to H^+ at the anode and reacts to O_2 at the cathode to generate H_2O_2 ³; iii) water is oxidized at the anode and reacts with O_2 that is reduced on a boron-doped cathode to generate H_2O_2 ⁴; c) For direct synthesis using a Pd membrane reactor, hydrogen atoms are produced from water electrolysis which occurs in a separate reactor compartment from the production of H_2O_2 ⁵. The hydrogen atoms move through the membrane where they react with O_2 to form H_2O_2 ; d) For indirect synthesis, anthraquinone is first hydrogenated using H_2 gas to form anthraquinol (AQ- H_2)⁶. Subsequent oxidation of

anthraquinol reforms the original anthraquinone in tandem with H₂O₂ formation. This process is widely used in industry; e) For electrochemical hydrogenation of anthraquinone to produce H₂O₂, anthraquinone is used as a redox mediator in an aqueous solution where it can transfer hydrogen produced from water electrolysis to O₂⁷. f) For indirect synthesis using the Pd membrane reactor, hydrogen is sourced from water electrolysis and separated from the AQ by the Pd membrane. This allows for efficient water electrolysis conditions in one compartment of the reactor while maintaining the organic solution in the AQ compartment of the reactor and allows for direct integration with current industrial methods for H₂O₂ production. Hydrogen gas denoted in red represents hydrogen sourced from steam methane reforming, while water and electricity denoted in green symbolizes that these can be sourced from renewable energy.”

- The introduction of the manuscript has been updated accordingly (pages 4–5 of the tracked document; pages 4–5 of the final document) – we have removed the extensive discussion and comparison to direct H₂O₂ production as it is now captured more appropriately in Supplementary Fig 1.
- We compare our direct H₂O₂ work to others.

Again, our work is **indirect** H₂O₂ production.

Reviewer 3 Comment 2

Supplementary figure 1 should be revised and you should do more literature study on the matter of electrochemical H₂O₂ formation. For example, your drawing of (b) is misleading

Our work features **indirect** H₂O₂ production, not direct H₂O₂ production. Notwithstanding, we have expanded Fig S1 to hopefully call out the appropriate literature for both indirect and direct H₂O₂ production.

REVIEWERS' COMMENTS

Reviewer #4 (Remarks to the Author):

I have carefully reread the manuscript and still aren't sure of few things.

Figure 2B is not a fair comparison since Pd membrane used in this work also act as a catalyst where other work did not use such catalyst. The use of expensive Pd may also contributed to the high hydrogenation rate (I agree with authors that your system is electrochemical where the hydrogen comes from water electrolysis.)

Supplementary Figure 7,8,11 are still hard to understand since x-axis is time and the results from the graph looks like it is a continuous measurement. Is it actually a continuous measurement where you can move your inlet to MS in between different compartment without interruption?

Goal of electrifying H₂O₂ synthesis is to use renewable electricity to generate useful chemical, H₂O₂. This can be achieved by either direct (Haotian Wang's) or indirect method (Yours). I strongly argue that you should compare yours indirect method to direct electrochemical method and compare pros and cons of different methods. I do value your indirect approach which is new and interesting. But as of now, indirect method seems inferior to direct electrochemical method... low current density, low current efficiency, complex system design, etc. Then can it be published in Nat. Comm.? I am not sure.

black (reviewers' comments) | blue (authors' response) | **bold** (manuscript changes)

REVIEWERS' COMMENTS

Reviewer #4 (Remarks to the Author):

I have carefully reread the manuscript and still aren't sure of few things.

Figure 2B is not a fair comparison since Pd membrane used in this work also act as a catalyst where other work did not use such catalyst. The use of expensive Pd may also contributed to the high hydrogenation rate (I agree with authors that your system is electrochemical where the hydrogen comes from water electrolysis.)

We have removed Figure 2B to avoid confusion and the potentially misleading comparison between our method and other electrochemical methods as raised by the reviewer.

Supplementary Figure 7,8,11 are still hard to understand since x-axis is time and the results from the graph looks like it is a continuous measurement. Is it actually a continuous measurement where you can move your inlet to MS in between different compartment without interruption?

Correct, the measurements are continuous. The atm-MS is running for the full duration of the experiment, even when the inlet is switched from the electrochemical to hydrogenation compartment. This occurs on the order of seconds, while the data collected is on the order of hours. Occasionally spikes in data are observed when the MS inlet is exposed to air for a longer period of time, as is evident with Fig S7 after AQ addition.

We have added this clarification to our experimental methods and to the figure caption to clarify our results for readers.

“The atm-MS ran and collected data continuously during the inlet changes between compartments. To the hydrogenation compartment was then added AQ until a 0.25 M concentration was reached and the hydrogen evolution was monitored in the compartment until a steady signal was recorded. Results were plotted over time only to show the stability of the signal. The inlet change during the experiment occurs on the order of seconds, while the data collected is on the order of hours.”

For Fig S11, no changes have been made as this is a continuous experiment in one compartment where the current density is increased over the duration of the experiment. The figure caption has been updated to clarify the experiment.

“Graph of ion current (left y-axis, blue) versus time and applied current density (right y-axis, orange) versus time **for a continuous electrolysis experiment measured in the**

hydrogenation compartment. The ionic current is proportional to H₂ permeation through the Pd foil membrane as measured by atmospheric–mass spectrometry.”

Goal of electrifying H₂O₂ synthesis is to use renewable electricity to generate useful chemical, H₂O₂. This can be achieved by either direct (Haotian Wang's) or indirect method (Yours). I strongly argue that you should compare yours indirect method to direct electrochemical method and compare pros and cons of different methods. I do value your indirect approach which is new and interesting. But as of now, indirect method seems inferior to direct electrochemical method... low current density, low current efficiency, complex system design, etc. Then can it be published in Nat. Comm.? I am not sure.

We now include Table S1 comparing indirect vs direct H₂O₂ production. Our method offers the best balance between current density, current efficiency, and H₂O₂ production without the use of H_{2(g)}. Moreover, our work can integrate with existing industrial H₂O₂ plants, which use the indirect method.

For your information, our method is superior to Haotian Wang's direct method (this table is an excerpt of the full table):

	Direct		Indirect
	Science 366, 226–231 (2019)	Nat. Commun. 12, 4225 (2021)	This work
H source	H _{2(g)}	H ₂ O	H ₂ O
Current density	200 mA cm ⁻²	400 mA cm ⁻²	≤150 mA cm ⁻²
Current efficiency	22.6%	85%	80±7% @75 mA cm ⁻² 42±4% @150 mA cm ⁻²
H₂O₂ production	6.29 g L ⁻¹	1.1 g L ⁻¹ @30 mA cm ⁻²	5.24 g L ⁻¹ @75 mA cm ⁻²